# Preserving Ethnoveterinary Medicine (EVM) along the Transhumance Routes in Southwestern Angola: Synergies between International Cooperation and Academic Research [note 1]

**DOI:** 10.3390/plants13050670

**Published:** 2024-02-28

**Authors:** David Solazzo, Maria Vittoria Moretti, José J. Tchamba, Marina Filomena Francisco Rafael, Matteo Tonini, Gelsomina Fico, Txaran Basterrecea, Silvano Levi, Lorenzo Marini, Piero Bruschi

**Affiliations:** 1FAO Angola Country Office, Largo Antonio Jacinto, 4° Andar, Luanda Caixa Postal 10043, Angolamatteo.tonini@unifi.it (M.T.); txaranb@yahoo.es (T.B.); 2Dipartimento di Scienze e Tecnologie Agrarie, Alimentari, Ambientali e Forestali (DAGRI), Università di Firenze, Piazzale delle Cascine 18, 50144 Firenze, Italy; mariavittoria.moretti@unifi.it (M.V.M.); lo.marini@unifi.it (L.M.); 3Centro de Estudos da Biodiversidade e Educação Ambiental, Instituto Superior de Ciências da Educação da Huíla (ISCED-Huíla), Rua Sarmeto Rodrigues, Lubango Caixa Postal 230, Angola; jose.tchamba@isced-huila.ed.ao (J.J.T.); marina.rafael@isced-huila.ed.ao (M.F.F.R.); 4Dipartimento di Scienze Farmaceutiche, Università degli Studi di Milano, Via Mangiagalli 25, 20133 Milano, Italy; gelsomina.fico@unimi.it; 5Instituto Superior Poletécnico Sinodal de Lubango, Rua cdt Hoji ya Henda, Lubango, Angola; sillevis@live.com.pt

**Keywords:** ethnoveterinary, EVM, transhumance, southwestern Angola, conventional veterinary medicine, decolonization

## Abstract

This study delves into the ethnoveterinary medicine (EVM) practiced by pastoralists along the transhumance routes in southwestern Angola. Within the framework of three cooperation projects, we conducted 434 interviews, collecting information on 89 taxa used for treating 16 livestock diseases. The most cited species was *Ptaeroxylon obliquum* (132 citations), followed by *Salvadora persica* (59) and *Elaeodendron transvaalense* (49). Contagious bovine pleuropneumonia (CBPP) was the disease most cited (223 citations; 44 species), followed by wounds (95; 20) and Newcastle (86; 14). We found that 30 species and 48 uses have not been previously reported in the ethnoveterinary literature. Jaccard index (mean value = 0.13) showed a greatly diversified knowledge among the ethnic groups: Kuvale and Nyaneka were the most knowledgeable and should be included in the various strategies for disseminating EVM in the area. Most informants recognized that abundance of some species decreased in the last years as a result of human activities and climatic changes. Finally, we discuss challenges in preserving the EVM in the area. Our findings suggest that preservation of the EVM in southwestern Angola is widely impacted by the access to biomedicine. Future studies should investigate the opportunity to integrate traditional medicine into mainstream development projects, which is crucial for decolonizing the veterinary sector in Angola.

## 1. Introduction

Pastoralism in southwestern Angola is a viable part of the local cultures where livestock herding not only plays a crucial role in ensuring livelihood and food security but is also the core of social life and culture as well as the base of religious and cultural identity of the indigenous communities [1]. The southwestern provinces have the largest number of livestock, accounting for 97% of total national bovine production; this region is characterized by erratic precipitations and high temperatures [2]. As a result of these climatic constraints, pastoralists relocate alongside their livestock on a daily and seasonal basis according to the distribution of fodder and water points [3] (Figure 1).

The rise of nomadism and transhumance in this region dates back to very ancient times [4]; for millennia, mobility allowed opportunistic use of key resources and helped in reducing the impact of drought on the livelihood of communities living in the area. Angola is now experiencing its most severe drought period in 40 years [5], and approximately 3.8 million individuals are facing inadequate availability of food in the southwestern provinces [6]. Drought and lack of resources forced thousands of people to cross the Namibia borders or to migrate to urban areas, causing the process of “human desertification” of the rural communities. Despite the past month’s increased precipitation, long-term rainfall shortage and deterioration of the vegetation biomass (20–40% below the average) persist in Huíla, Cunene, and, in particular, Namibe [7]. Climatic events have detrimental impacts on the availability and quality of pasture, on the accessibility of water points, and, consequently, on the production and survival of livestock. Cattle, in particular, have been shown to be more affected due to their feeding habits and susceptibility to heat stress [8]. In addition, drought exacerbates the effects of parasites and infectious diseases caused by reduced immunity associated with poor nutrition and the increased risk of transmission in overcrowding conditions near water and food sources. The introduction of public measures to encourage animal vaccinations and the use of pharmacological treatments have contributed significantly to the prevention and control of the most serious animal diseases in Angola [9]. However, several political and socioeconomic barriers continue to limit access to veterinary medicines in Angola and sub-Saharan Africa in general [10]. Angola, for example, is completely dependent on imports of veterinary medicines from other countries and lacks an effective regulatory system addressing veterinary drugs [10]. Another problem is related to the difficulty of organizing a network of infrastructures and services able to rapidly meet the needs of all farmers; poor infrastructure and the limited availability of cold chain equipment make it difficult to reach cattle heads in remote areas before the vaccine is degraded to the point that it cannot ensure immunity. All these limitations make it urgent to search for alternative and complementary approaches to heal livestock diseases. In the last years, there has been an increased focus on ethnoveterinary medicine (EVM) due to the scientific validation of the efficacy of some traditional herbal remedies as well the awareness that this precious knowledge is rapidly disappearing [11,12]. EVM has been mostly investigated in Africa, Asia, and North America [13]. Regarding the southern areas of Africa, most of the research is from South Africa (see, for example, the review by [11] and the papers by [14,15,16,17]). More sporadically, studies have been carried out in Zimbabwe [18,19,20,21,22,23], Botswana [24,25,26], Mozambique [27], Namibia [12,28], and Zambia [29]. To the best of our knowledge, only one paper has been published on the ethnoveterinary uses in Angola [30]. The research presented in this paper is the result of a partnership between international cooperation and the scientific community, within the framework of three different projects: PIRAN, RETESA, and FRESAN.

We studied ethnoveterinary medicine practiced by pastoralists along the transhumance routes (Figure 2) crossing the southwestern provinces of Angola in order to outline the importance of the local ecological knowledge (LEK) for improving animal health and to discuss the opportunities and challenges of incorporating traditional knowledge-based practices into projects of development assistance to veterinary sector. We think that a rights-based strategy must integrate and underscore the LEK as a pivotal element for both decision-making and understanding of the actual needs of the local communities. The specific objects of this paper are (1) to record veterinary plants and their uses; (2) to identify differences and similarities of ethnoveterinary uses among ethnic groups living in the area; (3) to get information on the conservation status of the plants mentioned by the informants; and (4) to discuss challenges and constraints of preserving the EVM in the studied area.

## 2. Results and Discussion

### 2.1. Informants

We interviewed a total of 434 pastoralists: 279 in the first phase of the fieldwork (2014–2017) and 155 in the second phase (2023) (Table 1; Figure 3). Three thousand and eleven people (72%) reported the veterinary use of one species. Out of the knowledgeable informants, 24 (7.7%) were from Benguela, 62 (20%) from Cunene, 66 (21.3%) from Huila, and 159 (51%) from Namibe (Table 1).

Most of them (54.7%) live in the dry steppes of the Namib savanna woodlands and Angolan Escarpment savanna ecoregions, 39% live in the dry woodlands of Angola Mopane ecoregion, and only 6% in the area of Angolan Miombo woodland ecoregion (see [2]) for the classification of Angolan ecoregions and vegetation types). Most of the informants were Nyaneka (21.5%), Kuvale (14.7%), and Muhumbi (14.5%) (Table 1); Kwanyama and Ovimbundu were other well-represented groups (10.2% and 9.6%, respectively (Table 1). Twenty-six informants did not report any information regarding their ethnicity. The observed distribution of informants by gender (91% males and 9% females) confirms the common pattern of greater male involvement in pastoral activities [12,17]. Livestock-related tasks, including herding and trading, are exclusively conducted by males, while females play crucial roles in milking cows, managing household chores, harvesting crops, and caring for children. Sixty participants did not know or remember their age; this is a common occurrence in rural areas of Angola [31]. The age of the other informants ranged between 17 and 88 (mean age = 46.9 ± 15.24) with the age group of 40–49 being the dominant one (22%), followed by the group of 50–59 (21%), and the group of 30–39 (19%). Thirty-five % of the interviewed people claimed to have received no education at all; the overall level of education is still poor and limited to the first grades of primary school.

### 2.2. Diversity of Veterinary Plant Species in South Angola

We identified 89 taxa belonging to 85 genera and 41 families for healing 16 different animal diseases (Table 2). Other eight identified species (*Boscia integrifolia* J.St.-Hil., *Brachystegia tamarindoides* Welw. ex Benth., *Croton mubango* Müll.Arg., *Entandrophragma angolense* C.DC., *Pittosporum viridiflorum* Sims, *Senegalia schweinfurthii* (Brenan and Exell) Seigler and Ebinger, *Solanum anomalum* Thonn., *Zanthoxylum leprieurii* Guill. and Perr.) have not been included in Table 2 because informants did not report the specific medicinal use. The number of species in our study is higher than reported in the same area [30] or in other Southern Africa regions [12,15,16,17,27]. This can be due to the greater number of informants interviewed or to the broader variety of ethnic groups involved in our study, which could have enabled us to cover a broader range of knowledge systems. Moreover, transhumance routes in southwestern Angola cross diverse ecosystems (dry steppes, *Miombo* and *Mopane*), and shepherds can have the chance to know and experience a high variety of plants. Plant species were compared with the available data from other ethnoveterinary studies carried out in other Southern Africa areas. The results showed low values of similarity (mean JI = 6.2 ± 1.7), confirming the richness and peculiarity of ethnoveterinary wisdom owned by the pastoralists of southwestern Angola. The highest value was found with studies conducted in Namibia [12,28] and Zimbabwe [18,19,20,21,22,23] (JI = 8.2), while the low values were obtained for Botswana [24,25,26] (JI = 3.5) and Zambia [29] (JI = 4.9). Comparison with South Africa [11,14,15,16,17] and Mozambique [27] showed a JI = 6.5 and a JI = 6.1, respectively. Most species (51%) had only one or two citations (Table 2) and only 20% of the species were cited more than ten times.

The most cited species was *Ptaeroxylon obliquum* (Thunb.) Radlk (Figure 4c) (132 citations; 17% of the total citations), followed by *Salvadora persica* L. (59; 7.7%), *Elaeodendron transvaalense* (Burtt Davy) R.H.Archer (49; 6.5%), *Capsicum frutescens* L. (47; 6.1%), *Aloe littoralis* Baker (Figure 5a) (46; 6%), and *Zygophyllum simplex* L. (Figure 5b) (43; 5.6%) (Table 2).

Most of the species belong to Fabaceae (22 species; 25%), followed by Asteraceae (6; 7%), Euphorbiaceae, Malvaceae, and Solanaceae (4 each; 4.5%). Similar findings were reported in the study conducted by [32] on the plants used for the human disease treatment in the same area. According to [33] Fabaceae and Asteraceae are the largest botanical families in Angola with 934 and 463 species, respectively; moreover, Fabaceae and Asteraceae have been reported to have several ethnopharmacological uses both in Angola and worldwide (see among others, [11,32,34]). Most of the Fabaceae mentioned in this study are important, or even dominant, species of the *Miombo* and *Mopane* ecosystems [34,35] where the interviewed shepherds live or that they cross during their seasonal migration. The floristic composition and physiognomic structure of these dry woodland ecosystems could also explain the results of the life forms analysis: perennial woody plants are the most numerous (70%; 54% trees and 16% shrubs), followed by herbs (21%, of which 17% are perennial), succulents (5.7%), and lianas (2.3%). Moreover, many studies [32,36] have shown that in strongly seasonal environments, woody plants are more frequently used than herbs. Trees and shrubs can supply a variety of useful products all year round, especially in the arid months, due to the biological and ecological characteristics making them more resistant to drought and other abiotic disturbances. As a confirmation of this, informants reported that 87% of the mentioned remedies (85% of all the citations) were available all year round, while 10% and 2% were available during rainfall and dry seasons, respectively. Among the 89 identified species, 74 are native and 15 are exotic. Most annual herbs were cultivated species that can be grown in gardens throughout the year, like *Solanum lycopersicon* L., *Nicotiana tabacum* L., and *Zea mays* L. Leaves were the most frequently quoted used parts (255 citations; 36% of the total citations). followed by roots (206; 29%), bark (158; 22%), and fruits (57; 8%). These results are consistent with those reported in many other ethnoveterinary studies [16,17]. The most frequent preparation method was maceration (379 citations; 50% of the total citations), followed by burn (71; 9.5%), paste (69; 9.2%) powder (63; 8%), infusion (48, 6%), and raw (40; 5%). The oral route was the main way of administration (467 citations; 62% of the total citations), followed by direct application (213; 28%), inhalation (56; 7%), and bath (19; 2.5%).

### 2.3. Diseases Treated by Plant Preparations

Plant remedies were mostly used for treating cattle (91% of the total number of citations; 84 species) and, in a secondary way, chicken diseases (7.9%; 14). Only a few citations were recorded for the veterinary uses concerning goats and pigs. The magnitude of this discrepancy confirms the importance that cattle herding plays in the economy and the sociocultural context of these people [37]. A man’s wealth depends on the number of heads he owns; cattle are an integral part of the life of each member of the community, and they are the main reason for physical and social mobility [38]. The economic use of cattle is mostly milk production, which is the basis of the agropastoralists’ diet and provides cash income during the year. As an example, the fermented buttermilk produced by Herero and Ovambo of northern Namibia and southern Angola plays an important cultural and economic role [39]. Poultry production in rural areas is a traditional household activity carried out mainly by women and children [40]. Caprines and ovines are not particularly important either culturally or economically, although they represent the currency of exchange and are consumed more frequently than bovine meat. Most ethnoveterinary uses (93%) had only one or two citations (Table 3) and only 2% of them were cited more than ten times. The higher number of citations (223 citations; 29% of the total number of citations; 44 species; see Figure 6b) regarded the treatment of CBPP (contagious bovine pleuropneumonia, locally known as *Cawenha*) (Table 3). This finding is in accordance with [30] and reflects the epidemiological situation of the area. An investigation carried out by [41] showed that 64% of farmers interviewed in Namibe identified CBPP as a primary factor of cattle death and estimated the prevalence of the disease to be more than 94%. More recently, ref. [42] found that 83% of total rejections in Benguela slaughterhouse were caused by the manifestation of CBPP symptoms. The most cited species for treating CBPP were *Salvadora persica* (54 citations; FL = 98%), *Ptaeroxylon obliquum* (50; 48%), *Spirostachys africana* Sond. (17; 81%), *Oldfieldia dactylophylla* (Welw. ex Oliv.) J. Léonard (12; 100%), *Peltophorum africanum* Sond. (11; 61%), and *Cissus quadrangularis* L. (10; 91%). The only literature references supporting this veterinary use were for *P. obliquum* [30] and *C. quadrangularis* [43]; however, extracts of *S. persica* [12,44], *P. obliquum* [45,46], *S. africana* [47], and *P. africanum* [48] have been shown to have noticeable antimicrobial activity. O. *dactylophylla* is reported as a remedy for pneumonia in humans [49]. Wounds were the second most cited disease category (95 citations; 12.5% of the total number of citations; 20 species) (Table 3) and included sores, abscesses, warts, and inflamed skin lesions. According to the informants, wounds are mainly caused by tick bites and thorny bushes or other plant materials.

The main plant used for treating wounds was *Elaeodendron transvaalense* (49 citations; FL = 100%), followed by *Peltophorum africanum* (7; 39%) and *Erlangea misera* S. Moore (6; 100%). Ethnoveterinary uses of *E. transvaalense* reported in the literature concern the treatment of diarrhea [50,51] and worms [51]. However, this species is traditionally used in human medicine to treat skin rashes and wounds [52] and as an emetic [53]. Several studies have demonstrated that *E. transvaalense* extracts have antibacterial and anti-inflammatory activities, thereby supporting their traditional uses [54]. *P. africanum* is used to treat wounds in humans [48]. No literature information is available on *E. misera* as a veterinary remedy. The only medicinal use reported in the literature for this species is related to vomiting and fever relief in Kwanyama children [55]. Newcastle was the only disease mentioned for poultry (86 citations; 11% of the total number of citations; 14 species) (Table 3). Newcastle is the major cause of chicken mortality in rural villages and represents an important constraint to traditional poultry production in Angola [40]. *Capsicum frutescens* (46 citations; FL = 98%) was the main drug used to treat Newcastle, followed by *Aloe littoralis* (12; 32%) and *Ptaeroxylon obliquum* (9; 9%). Remedies against Newcastle, based on fruits of *C. frutescens* and leaf sap of *Aloe* spp., have been reported in studies conducted in other African countries [23,56,57,58]. While no prophylactic or therapeutic value of *C. frutescens* extract against Newcastle virus has been found [59], a reduced mortality rate has been observed in chickens treated with *Aloe secundiflora* Engl. leaf extract [60]. Other frequently mentioned treatments concerned scabies (83 citations; 10.8% of the total number of citations; 16 species), viral and bacterial diarrhea (60; 7.8%; 14), symptomatic carbuncle (46; 6%; 17), snake or scorpion bites (41; 5.4%; 12), and LSD (lumpy skin disease) (40; 5.2%; 12). The most frequently cited plant against scabies was *Zygophyllum simplex* (43 citations; FL = 100%), followed by *Securidaca longipedunculata* Fresen. (9; 69%), *Otomeria elatior* (A.Rich.) Verdc. (7; 54%), and *Ximenia americana* L. in Figure 4d (6; 75%). *Z. simplex* is reported by Cholistan pastoralists to treat livestock skin problems [61], but it is also documented as a traditional remedy for the treatment of dry scaly patches and cleaning the skin in humans [62]. *S. longipedunculata* has been proven to have high pesticidal effects against several insect species [63]. *X. americana* is used to treat internal parasites in cows and has been proven to have acaricide activity against the cattle parasite *Rhipicephalus microplus* [64]. In southern Angola, an oil obtained from the kernels of this species is widely employed to cure skin problems in humans. *P. obliquum* was the most reported species to treat diarrhea (28 citations; FL = 27%), followed by *P. angolensis* (6; 100%). *P. obliquum* is used against intestinal parasites in goats and, also, to treat diarrhea in humans [32]. *P. angolensis* is reported to have anthelmintic, antibacterial, and anti-inflammatory activities [65,66]. *P. obliquum* was also the most used species to treat symptomatic carbuncle affection (or anthrax) (16 citations; FL = 15%). This is in accordance with what was reported by [67] and [45]. The main species against snake or scorpion bites were *Senna occidentalis* (L.) Link (15 citations; FL = 75%) and *Sclerocarya birrea* Hochst. (11; 55%). *S. occidentalis* is widely employed in traditional medicine for the treatment of local tissue damages in humans induced by snake and insect bites [68]. Ref. [32] reported the same use in the Namibe area. *S. birrea* is known as a remedy in human medicine for the treatment of snake bites in Senegal [69], Tanzania [70], and Ghana [71]. *P. obliquum* (12 citations; FL = 11%) was the most frequently cited species for treating LSD, followed by *Spirostachys africana* (7; 33%) and *Peltophorum africanum* (5; 28%). Other interesting and new ethnoveterinary uses were recorded for the treatment of conjunctivitis with the endemic species *Thesium cinereum* A. W. Hill (16 citations; FL = 100%) and *Cenchrus americanus* (L.) Morrone R. Br. (6; 100%). The only traditional medicinal use of *T. cinereum* reported in the literature is for the treatment of human bronchitis in southern Angola [72]. Other species belonging to the genus *Thesium* are used to enhance the sight [73]. Ref. [16] report *Thesium* sp. in the treatment of diarrhea and internal parasites in calves and cows. *C, americanus* is used to heal brucellosis in cattle. However, a similar species, *Pennisetum pedicellatum* Trin., is used to treat human eye infections in northern Nigeria [74]. The most versatile species was *P. obliquum* (11 uses), followed by *Aloe littoralis* (9), *S. longipedunculata* (6), *P. africanum*, *S. occidentalis*, and *S. africana* (5 each). By comparing our research results with previous studies, we found that 30 species (58% of the total number of species) and 48 uses (67% of uses recorded for the already known species) have not been previously reported in the ethnoveterinary literature (Table A2 in Appendix B). These findings show that our study provides an important contribution to the ethnoveterinary medicine of southern regions of Africa and, also, highlights the importance of recording new ethnobotanical information for already known medicinal plants.

### 2.4. Plants and Uses among Different Ethnic Groups

There were 63% of plants and 58% of uses recorded from at least two ethnic groups. *Ptaeroxylon obliquum* was reported by 10 ethnic groups; *Aloe littoralis*, *Capsicum frutescens*, *Elaeodendron transvaalense*, *Peltophorum africanum*, *Salvadora persica,* and *Zygophyllum simplex* were reported by 6 ethnic groups; and *Cissus quadrangularis* and *Spirostachys Africana,* by 5 ethnic groups. *P. obliquum* against CBPP was the most common throughout different ethnic groups (8 ethnic groups); *C. frutescens* to treat Newcastle, *E. transvaalense* to heal wounds, *P. obliquum* to treat diarrhea, *S. persica* to treat CBPP, and *Z. simplex* to treat scabies were reported by 6 ethnic groups. *C. quadrangularis* against CBPP, *P. obliquum* against symptomatic carbuncle, and *S. africana* against CBPP were reported by 5 ethnic groups. The high diversity of ethnoveterinary knowledge existing in the area was confirmed by the low values of the Jaccard similarity index (JI mean value = 0.13 ± 0.12). The highest value was 0.50 when comparing Mumwila (Nyaneka-Humbi ethnolinguistic group) and Himba (Herero ethnolinguistic group). Muhanda (Nyaneka-Humbi ethnolinguistic group) was the most different with JI values ranging from 0.0 to 0.09. Ovambo and Kwanyama, belonging to the Ambo ethnolinguistic group, showed a certain similarity (JI = 0.44) and were strongly different from the other ethnic groups. Kuvale and Nyaneka were particularly skilled in treating CBPP, Newcastle, scabies, LSD, and cough (Figure 6). Kuvale also had the higher number of unique uses (6), followed by Muhanda (4) and Nyaneka (3).

These results suggest that Kuvale and Nyaneka should be included in the various strategies for promoting and disseminating knowledge about the use of ethnoveterinary plants. Kuvale has developed a lot of knowledge about the rearing and management of livestock. Although their society is greatly conservative, it is possible to find some similarities with other groups living in the same area like the Himba, the Nyaneka, the Mwila, and the Muhimbi [75]. Similarities with these groups can also be found in the use of veterinary plants, suggesting an exchange of knowledge and experiences on the treatment of animal diseases along the transhumance routes. On the other hand, the same linguistic ancestry characterizing all the Bantu languages can provide important points of convergence between ethnic groups belonging to the Nyaneka-Humbi ethnolinguistic group; this also true for Kuvale and Himba that are believed to speak the same Herero dialect, which, along with Nyaneka-Humbi languages, has been included in a subcategory of Southwest Bantu languages [76].

### 2.5. Conservation Status of the Mentioned Species

Most interviewees (78%) recognized that the abundance of some species (Figure 7) (e.g., *Elaeodendron transvaalense*, *Ptaeroxylon obliquum*, *Salvadora persica*, *Zygophyllum simplex*) decreased in the last years and that these changes are mainly related to human activities and climatic changes. Distribution of abundance categories based on informants’ perception showed that 202 (42%) citations accounted for high abundance, 150 (31%) for medium abundance, and 131 (27%) for low abundance (see Table A1 in Appendix A; Figure 7). Only 16 (18%) mentioned species were reported in an inventory study conducted by [77] on Mopane vegetation in the same area (Table A1 in Appendix A). Species having higher density in the inventory plots were *Colophospermum mopane* (J.Kirk ex Benth.) (PAI: 2.7), *Croton gratissimus* var. *gratissimus* (2.4), *Ximenia americana* (2.3)*, Salvadora persica* (2.0), and *Spirostachys africana* (2.8). Rarely encountered species were *Peltophorum africanum* (PAI: 1.9), *Sclerocarya birrea* (1.7), and *Elaeodendron transvaalense* (1.7). *Ptaeroxylon obliquum*, which was the species most cited by our informants, had an abundance of about 13 individuals/ha (PAI: 2.2). Similar density values have been found by [78].

Thirty-eight wild species mentioned in our study are included in the Red List [79] (see Table A1 in Appendix A); all of them are reported as the least concern (LC) and stable with the exception of *Philenoptera pallescens* (Welw. ex Baker) Schrire, which is categorized as vulnerable (VU) and decreasing. Ref. [80] reported fifteen species as VU, one as EN (*Diospyros mespiliformis* Hochst. ex A.DC.), and five as LC. Among the most cited species in our study, six are classified as VU by these authors (see Table A1 in Appendix A). The analysis of endemism showed that four species (*Dicoma antunesii* O. Hoffm., *Helichrysum benguellense* Hiern, *Philenoptera pallescens*, *Thesium cinereum*) are endemic to Angola.

The relatively restricted distribution of these species makes them potentially more susceptible to environmental threats. Knowledge of perceptions can be used to establish baselines in the first phases of a conservation project and to have a rapid overview of the conservation status of species when other data are not available [81]. However, local perceptions can be greatly subjective and do not always represent the actual status of the environment [82]; this can explain the lack of a significant correlation (R = 0.17; *p* > 0.05) between PAI and density values in the inventory studies. Despite experiencing the same ecosystems, different cultural backgrounds may have different views on the conservation status of ecosystems. In the case of *Securidaca longipeduncolata*, Nhanyeka reported the species as rare, while Kuvale indicated it as highly abundant. *Ptaeroxylon obliquum* was abundant according to Kuvale, Muanha, and Muhanda perceptions, and as rare according to Nyaneka. *Oldfieldia dactylophilla* was reported as abundant by Kuvale and as medium by Muhumbi. When plants were mentioned only by one group, we often observed absolute concordance in terms of abundance categories. For example, *Cenchrus americanus* and *Pterocarpus angolensis* (Figure 4a) were mentioned only by Muhanda and all the six informants reported these species as abundant in the area. The same concordance pattern was observed for *Thesium cinereum* and *Cassia angolensis* Hiern. that were reported by Kuvale (*T. cinereum*) and Nyaneka (*C. angolensis*) as abundant and medium, respectively.

### 2.6. Folk Names of Reported Plants

The scientific identification of plants is a process demanding a considerable amount of time and it requires well-trained and highly skilled professionals. In this regard, a repertoire of folk names with the corresponding scientific names may be a very useful tool for a rapid assessment of resource uses in remote areas, especially during Rapid Rural Appraisal (RRA) surveys [83]. However, while recognizing the significant role of folk nomenclature in providing valuable insights on LEK, many authors have pointed out the risk of misidentification and misinterpretation by an ethnobotanical approach based only on the collection of folk names [84,85]. In our study, 149 folk names were recorded for the 89 identified taxa. Fifty-four percent of these folk names matched those included by [86] in their list and could be ascribed to the same species; however, when comparison was made with data reported in [30,32], we found that only 9 folk names could be consistently equated with the 18 species common to these datasets. These results suggest that folk names could provide a valuable contribution to the rapid identification of the ethnospecies. At the same time, a critical approach when using this type of data for ethnobotanical inventory purposes is crucial. We observed that some folk names had correspondence with two or more species in [86]. For example, the Nyaneka name “onambumbu”, identified as *Croton gratissimus* var. *gratissimus* in our study, is ascribed by [86] both to *Croton gratissimus* var. *gratissimus* and *Croton integrifolius* Pax.; the Mumuhuila name “katete”, identified as *C. integrifolia* in our study, was reported both as *C. integrifolia* and *Eugenia malangensis* (O. Hoffm.) Nied. in [86]. Thirty percent of folk names were not reported at all by [86] and thirteen percent were reported with a slightly different transcription (e.g., “chifuaca” vs. “tchifuica” = *Fadogia cienkowski* var. *lanceolata* Robyns; “muipanhoca” vs. “murianhoca” = *Senna occidentalis*; “nganga” vs. “unganga” = *Burkea africana* Hook.; “opandambala” vs. “omupndambale” = *Philenoptera pallescens*; “opukange” vs. “omupange” = *Dichrostachys cinerea* (L.) Wight and Arn.). Last, 3% of the folk names were ascribed to different species in [86]. For example, the Umbundu name “ongombe” that we identified as *Commiphora mollis* (Oliv.) Engl. was ascribed by [86] to three different species: *Argyrolobium aequinoctiale* Welw. ex Baker, *Cordia sinensis* Lam., and *Rothmannia engleriana* var. *ternifolia* (Ficalho and Hiern) Somers. These incongruences could be the result of linguistic issues [72,86] or, in some cases, of errors in transcribing the information by the different interviewers involved in this study (e.g., the names “Lumbungululu”, “Lumungululu”, “Mbungululu”, “Mbungulolo”, “Mumbungululu” reported during the interviewers are all spelling variations of the same folk Nyanheca name for *Ptaeroxylon obliquum*) (Figure 4c). It is undeniable that the risk of confusion is particularly high in a country like Angola where a multitude of different languages is spoken [87] and local variants can be found within the same linguistic group; in this regard, familiarity of the interviewer with the local languages is a factor strongly affecting the process of data collection. Moreover, morphological similarities and ecological convergence can amplify the risks associated with the abundance and variety of folk names. The use of folk names remains crucial for communicating with local people and public institutions about the importance of preserving LEK and can provide guidance in identifying a taxon when no other information is available. However, our results indicate that each record should be supported by a voucher specimen and validated through identification by an expert on the local flora.

### 2.7. Challenges and Constraints along Transhumance Routes

The practice of transhumance has changed over the years due to environmental, economic, and social factors, as reported by the informants. The main problems affecting the availability of resources in the region are habitat destruction due to timber harvesting/charcoal production and overgrazing by livestock [88]. Since 2008, the south of Angola has been repeatedly hit by droughts and floods, affecting the quality of pastures and the health of livestock. This situation is expected to worsen in the coming years. Average annual temperatures are projected to warm by 2.0 and 3.0 °C between 2040 and 2060 [89]. The impact of these changes on people’s livelihoods is already being felt, particularly in terms of water availability and vegetation cover. Participants in our study largely agreed that the decline in rainfall and the unpredictable and shortened rainy season are leading to water scarcity and a reduction in grazing land in the region. This view is exemplified by the following quotations: “*It’s been 10 years without rain, water and pasture are now rare*” and “*Now it does not rain anymore and we are forced to walk livestock over long distances*”. In addition, land grabbing and the occupation of common land, which includes trails connecting seasonal grazing areas, are fuelling landscape fragmentation and threatening the survival of transhumance routes [90]. A shepherd from Tombwa told us: “*Many losses on the road, as often the animal doesn’t make it and die*” This refers to the exhaustion of the animals due to fatigue and high temperatures and the struggle for grazing and water along the transhumance routes. Some informants also reported that attitudes toward mutual aid and solidarity, which have historically played a key role in the social cohesion and resilience of pastoralist communities [91], have recently declined due to climate change and land rights issues. Disputes and conflicts over grazing land and water points often occur along the border areas. Surveys revealed that conflict-prone areas also include regions close to provincial and municipal borders where overlapping resource claims lead to disputes. Frequent areas of conflict are the main destinations of transhumance, i.e., Pediba, Quilemba Velha, Lola, and Mutipa, where herd movements overlap with sedentary communities and their agricultural activities. Conflicts are more common among young people, who are less willing to engage in dialogue and respect rules and traditions and are often exacerbated by ethnic differences. Conflict resolution mechanisms include community meetings convened by the traditional authorities. Resolutions usually involve compensatory measures, such as the payment of livestock or financial compensation, to redress grievances and restore harmony between communities and/or pastoralists. According to interviewees, adaptation strategies that emphasize dialogue, cooperation, and negotiation between pastoralists and local communities, always with the involvement of traditional authorities and leaders, have been used in response to these changes. Proposals to adapt to the changing climatic conditions include creating water reserves, planting more fodder trees for livestock, and expanding grazing areas. Preserving cultural aspects remains a priority, but adaptations could ensure the sustainability of practices in the face of changing circumstances. The resilience of the community and its ability to overcome challenges through dialogue and collaboration are the key themes of the analysis.

### 2.8. Western Veterinary Medicine and Indigenous Practices

Our findings suggest that the management of livestock diseases in southwestern Angola is widely influenced by the access of shepherds to biomedicine. The recent promotion of free mass vaccination campaigns in Angola and the efforts made by the Angolan authorities to make the veterinary service more efficient throughout the whole territory have significantly improved the prevention and containment of CBPP, hematic and symptomatic carbuncle, and LSD [9]. In addition to causing livestock death, lack of regular vaccination and parasite control can contribute to organs and carcass rejections in slaughterhouses [42]; this severely impacts livestock productivity and, therefore, farmers’ economic welfare. Then, it is unsurprising that most of the respondents claimed to currently rely on Western veterinary medicine rather than on traditional plant remedies. A Nanyeka informant stated that “*if a cow does get sick, we call the doctor*”. A Kuvale woman said: “*We buy medicines in the animal store*”. Another informant: “*We use conventional medicine because it works well*”. For these and other shepherds stating similar assertions, the use of traditional EVM was apparently of no use. They believe that the government veterinary service has better expertise for the prevention, detection, and management of animal illnesses and that Western medicine is more effective than traditional practices. This perception is transversal to all ethnic groups and can be presumed to be basically independent of any cultural membership. Some studies have shown that farmers are pragmatic and evaluate/absorb information in a practical and utilitarian way regardless of whether it comes from Western science or any other source [92,93]. When a particular piece of knowledge proves to be useful, economically sound, and socioculturally acceptable, it will be adopted [94]. A predominance of livestock treatments through conventional medicine was also observed by [30,95] in the same provinces where we conducted this study. Conversely, refs. [16,96] showed that in South Africa, most indigenous people preferred to use traditional approaches where Western medicine was available. As suggested by [97], the hegemony of the Western approach can have a strong impact on the conservation of indigenous health traditions, but at the same time, it can cause the opposite effect of cultural resistance and revalorization of traditional medicine. Some herders interviewed in our study seemed to be reluctant toward the adoption of Western medicine, expressing trust in the traditional approach. This perception is exemplified by the following quotation: “*Plant remedies work well for me*; *there were times we had only plants to heal animals. Now there are vaccines but I continue to use plants because they are effective*”. This resistance to Western science could be due to deep-rooted beliefs transmitted within the clan or the community from generation to generation: “*Our forefathers used plants, our fathers used plants, we continue to use plants*”; “*elders of the community passed on their knowledge to the younger ones, and that is how it works to this day*”(Figure 8a).

Responses from other informants suggest that government veterinary procedures are sometimes viewed with suspicion by some shepherds; this lack of trust in institutions could have led to the resistance of people to the acceptance of veterinary services and to the adoption of chemicals to treat livestock diseases. One shepherd mentioned that plant medicines are healthier whereas pharmaceuticals have a negative impact, suggesting that another possible constraint to the use of Western medicine is the lack of notions about how to manage and apply veterinary medicines to livestock. An uncorrected utilization can produce counterproductive effects on the animal health causing distrust in conventional products. These adverse impacts can be exacerbated by illegal veterinary medicines, including falsified, unregistered/unauthorized products with low active molecule concentration or low bioavailability [98].

### 2.9. Importance of Preserving EVM

It is conceivable that the introduction of Western practices might have impacted the traditional understanding of livestock health care and on the conservation of ethnoveterinary knowledge in the studied area. In a similar way, ref. [99] suggests that the development of veterinary services promoted by state subsidies would have caused the abandonment of EVM in Russian Karelia. Ref. [100] supposes that modernization of veterinary practices in Kenya relegated EVM to a subordinate position where traditional approaches are considered primitive and anachronistic. Two Kuvale shepherds encountered in 2023 said to us: “*What do we need plants for when vaccines are available? Our elders know the uses of healing plants but we do not need to know them*”. Other informants reported that many more plants were used in the past to treat livestock diseases. By comparing data from the two recording periods, we observed that the informants quoting almost one veterinary species were 82% in 2014–2017 and 51% in 2023. A great reduction in the number of reported species and uses was also observed: species were 79 in 2014–2017 and 24 in 2023; uses were 148 in 2014–2017 and 52 in 2023. In particular, this reduction is associated with the disappearance of plants against internal parasites (8 species in 2014–2017) and LSD (12), and a lower number of species for treating carbuncle (14 in 2014–2017 vs. 4 in 2023), Newcastle (13 vs. 3) and wounds (20 vs. 1). Even taking into account the limitations due to design effects (i.e., different number of informants interviewed and data collected by different interviewers), these findings raise concerns on the conservation of EVM in the studied area. The risk of losing traditional practices does not regard only the healing practices; 42% of the interviews also revealed that cultural items traditionally associated with transhumance, such as ritualistic sacrifices and community celebrations before departure, have vanished in some areas. In this regard, an informant said that “*in the past, a goat was killed before leaving with the cattle, now they no longer do that, they just take the cattle and go*” (Figure 8b). Preserving and valorizing this piece of indigenous knowledge is crucial for ensuring the resilience of the pastoral communities coping with harsh environmental conditions like those found in southern Angola. EVM might provide valuable solutions when addressing minor health issues where the cost of veterinary services is unaffordable and where, especially in remote communities, access is limited [101]. Angola is dependent on imports of veterinary medicines from other countries [10] Medicine shortage makes it hard to meet the medical needs of local communities, so some livestock diseases cannot be treated according to the canons of Western medicine. For example, the country has experienced five years of consecutive economic recession since 2016, which led to a discontinuity in the vaccination rate and to the spread of some epidemic diseases causing high production losses [42]. Ref. [102] reports that Newcastle vaccinations, even if available since 2012, have not reached remote villages of southwestern regions of the country. Another important issue to be considered is the high and often unaffordable cost of medicines not covered by the National Veterinary Service. The cost of antibiotics used for healing wounds ranges from AOA 1550 to AOA 18,600. An acaricide may cost AOA 19,700. As a comparison, the selling price of milk ranges between AOA 200 and AOA 400 per liter; the market price of cattle ranges between AOA 150,000 and AOA 300,000, depending on the body weight and quality. This practice not only increases the costs associated with the vaccination but also exposes herders to the risk of theft and robbery while traveling. Similar constraints to the use of orthodox livestock medicines have been underlined by [103] in South Africa. The need to travel to a veterinary pharmacy, which is often far away, means additional transport costs, which worsens the herder’s economy [104]. Another motivation for preserving EVM knowledge is related to the need for alternatives to conventional medicines. The inappropriate and repeated use of antibiotics and acaricides is known to increase the number of drug-resistant pathogens, the accumulation of chemical residues in the food chain, and negative environmental effects. The use of plant remedies can contribute to the reduction of this problem; plants have a complex and refined mix of different active biomolecules able to cover a wide spectrum of therapeutic indications. In this regard, ref. [105] affirms the need to find substitutive treatments for the control of endo- and ectoparasites in Angolan livestock by using bioactive plants. Moreover, due to their easier and more rapid metabolization rate, phytochemicals reduce the risk of contaminants in meat and dairy products [106].

### 2.10. EVM in the Framework of Development Projects

Some papers [107,108] stress the importance of making the interests and requests of the academy match those of people living in the studied communities. This concept encompasses different knowledge domains and implies a continuous bidirectional exchange of information between professionals/researchers and local actors, with mutual learning opportunities for mutual benefits. Ref. [109] suggests that recording information on indigenous veterinary practices is the most positive way to face problems of tropical animal health within the framework of assistance programs. Ref. [100] stresses the importance of a bottom-up approach where the indigenous perception of livestock health problems plays a crucial role in planning and developing projects, making interventions more adapted to the sociocultural context in which people live and work. How to integrate indigenous knowledge into the practice of the developmental process is the real challenge. According to [11], both modern medicine and EVM have their advantages and limitations. As already pointed out, the application of Western medicine has several constraints in African countries; on the other hand, plants cannot eradicate epidemics like CBPP or carbuncle and do not serve as a universal solution for all animal diseases. This awareness was also shared by shepherds interviewed in this research, who mostly claimed to rely on conventional Western medicine. Preserving traditional culture remains a priority, but calibrations are needed to allow indigenous communities to cope with the ever-changing environmental and socioeconomic conditions. For this reason, focusing on EVM as a radical alternative to the Western veterinary health care system is not a viable option for local communities. As observed by [94], a representation of indigenous knowledge as something untainted to be preserved against the progress of Western science is misleading because it creates an image of the indigenous culture as conservative and resistant to innovation. Moreover, the view that indigenous knowledge produces a more reliable and sustainable solution to the problems of local communities than Western practices is illogical and dangerous. The crucial action to take to preserve EVM in African countries is to negotiate a way of integrating traditional medicine into mainstream development projects and health policies. This strategy could allow curbing the erosion of ethnoveterinary knowledge and fostering decolonization of the veterinary and paraveterinary sectors. Difficulties in bridging the gap between indigenous and Western scientific paradigms have been widely discussed in the literature [110,111]. However, some examples of collaborative initiatives concerning the human health care system demonstrate that the dichotomy between Western veterinary medicine and African healing methods can be overcome, allowing for the development of hybrid knowledge that evolves and is continually redefined [111,112]. Ref. [110] suggests working at the interface, identifying convergence points between the two paradigms, and working jointly toward the same health goal. Exploring EVM practices, as we have accomplished in this study, can be considered a first step in this approach because it helps to identify the problems regarding the management of livestock health on the ground and provides scientific information on therapeutic solutions based on indigenous perception and values reflecting local observations and knowledge system. Western science can supply tools and expertise functional to the validation and standardization of traditional procedures. Working within the two health systems avoiding a Eurocentric approach or, on the contrary, a romanticized view of traditional practices allows for a mutual acknowledging of the epistemologies upon which these systems are founded [111]. It is crucial to underline that the role of research, in this perspective, is to allow for a better contextualization and integration of reality, not supplanting it, and as highlighted by [113], the reality is more complex than drawn by a laboratory essay. The efficacy of a plant extract must be validated taking into account not only the synergistic action of different molecules and its effect on the organism in situ but also the specific economic, sociocultural, and spiritual context where the knowledge has been produced and where the reworked knowledge will be brought back. Ethnobiology, as a multidisciplinary science operating at the interface between biology and culture, plays a pivotal role in mediation by promoting mutual understanding between diverse knowledge systems and developing strategies to legitimate each other’s way of viewing animal health care. Ultimately, this can lead to enhanced health outcomes for livestock within the framework of assistance projects implemented in culturally different indigenous communities.

## 3. Conclusions

In our paper, the use of 89 taxa has been reported and a valuable number of new ethnoveterinary species and uses have been detected, which must be added to the current repertoire of veterinary plant uses recorded in the Southern African regions. This means that EVM is an important part of the local cultural heritage and can significantly contribute to the resilience of the local livestock-oriented livelihood systems. On the other hand, our findings suggest that the preservation of EVM in southwestern Angola risks being negatively impacted by access to biomedicine. Information collected and discussed in this paper can be a starting point for building a syncretic framework, in which diverse veterinary knowledge systems are allowed to coexist in different ways. The rather recent Angolan law (Presidential Decree No. 253/20) that recognizes the importance of traditional medicine as a tool to be incorporated into the national health system in Angola can be a further encouraging step. Further research is needed for a deeper understanding of the pharmacological activity, therapeutic use, and possible toxic effects of the less-known mentioned species. We want to conclude this paper by highlighting that medical pluralism has been a characteristic feature of Atlantic African societies (and Angola was not an exception) even before the European settlement [114]. The arrival of colonists did nothing more than add new medicinal knowledge to this plurality. As observed by [114], cross-cultural interaction and mutual acceptance of medical practices were still the norm in early modern times; Africans and Europeans were open to trying out new healing practices and new drugs, both perceiving these interactions as chances to learn new skills. Now, we should restart from this “innocence age” where a melting pot of different cultures permeated the health systems of African colonies before Europeans, seduced by the ideas of their cultural supremacy, began to deny traditional knowledge.

## 4. Material and Methods

### 4.1. Study Area

The study area covers 15 municipalities in 4 provinces in the southwest of Angola: namely, Namibe, Huila, Cunene, and Benguela. (Figure 2 and Table 1).

The region, commonly referred to as *Sudoeste Angolano* [115], is bounded by the Atlantic Ocean in the west and the Marginal Mountain Range of Huila and Cunene in the east; it borders Namibia in the south and is delimited by the Coporolo intermittent river in Benguela Province in the north [116]. According to an estimate by the Angolan Institute of Statistics (INE), the Province of Namibe is home to approximately 672,086 inhabitants [117], Cunene hosts approximately 990,087 inhabitants, Huila 2,497,422, and Benguela 2,231,385 [118]. The study area shows a large variety of climatic conditions: from the sub-humid zones affected by the Serra da Chela Escarpment and the Humpata Plateau’s influence [119] to the hyper-arid desertic climate of the south part of Namibe Province. The area experiences the impact of the interplay between the cold Benguela current, the Namib Desert, and the elevated terrain [2]. The mean annual temperature is 20 °C and the mean annual precipitation is 37 mm [2]. The rainy hot season occurs from January to March, while the long dry season lasts about nine months [2].

### 4.2. Ethnic Groups Living in the Area

People living in the area belong to 4 main ethnolinguistic groups and approximately 12 distinct ethnic groups (*povos* according to [87]). The Nyaneka-Humbi ethnolinguistic group includes the following ethnic groups: Muhanda, Muhumbi, Mumuhuila, Mungambwe, and Muquilengue. They are mainly settled in the eastern foothills of the Chela mountain range [120] and have a semi-nomadic lifestyle, mostly practicing subsistence agriculture while also keeping livestock such as poultry, goats, and cattle. They are widely known for their beekeeping practices and honey production. The Herero group comprises Kuvale (also known as Mucubal), Himba, Muacahona, Mundimba, Tjimba (Cimba), Mbanderu, and Kwandu. Herero are widespread in Southern Africa and live mainly in Namibia, while smaller communities can also be found in Botswana and southern Angola. Unlike most Bantu groups, the Herero are traditionally nomadic pastoralists whose main occupation is herding cattle and whose social status is determined by the number of animals they own. The Herero have mastered the art of food preservation, using techniques such as souring milk and drying meat to extend the shelf life of these perishable foods. Ovimbundu/Umbundu live in the Benguela highlands and the wooded savannas along the coastal areas west of these highlands. They are considered the largest ethnic group in Angola (40% of the whole population). Ovimbundu are mostly farmers, with a notable contribution from women. Livestock rearing involves cattle, goats, pigs, and poultry, with cattle being an important form of investment. Traditional activities such as fishing, hunting, and trapping are practiced during the dry season, reflecting the diverse livelihood strategies within this community. The Ambó ethnolinguistic group, also known as the Ovambo, comprises various subgroups, including the Kwanyama ethnic groups, who live in the flat sandy grass plains of Cunene Province [121]. They lead a sedentary life and live mainly on a combination of agriculture and livestock rearing, cultivating staple crops such as millet, sorghum, and beans. In drier seasons, livestock farming with herds of cattle, goats, and sheep plays an important role as a source of milk, contributing to the livelihood of people. The Ambó are also skilled artisans who make and sell baskets, pottery, jewelry, wooden combs, wooden iron spears, arrows, and other items.

### 4.3. Transhumance Practices

All ethnic groups interviewed (Table 1) are involved in transhumance; in particular, Kuvale, Nyaneka, and Umbundu are known to have a deeper knowledge and experience in traditional transhumance practices. The movement of cattle spans several municipalities within key areas (Figure 2), including breaks for cattle about every 15 km. The transhumance locations are generally the same from one time to another, although the routes can undergo modifications depending on the annual availability of pasture and water points. The cattle are often brought to specific grazing grounds, such as vast plains, foothills, and riverbanks, providing a diverse range of vegetation for the animals and watering resources. The timing related to departures and returns of the herds varies depending on the needs of the shepherds, the climatic events (i.e., timing and length of rainfall events), and the areas of departure.

### 4.4. Data Collection

The ethnobotanical investigation was conducted in collaboration with the Veterinary Services Institute (ISV) and the Herbarium of ISCED (Instituto Superior de Ciências de Educação). The study combines two surveys carried out in different periods but in the same geographical areas during the migration along the main routes. In the first survey, data collection was conducted by the technicians of lSV from 2014 to 2017.

The second phase of data collection was carried out by ISCED personnel in 2023. Informants were selected using snowball sampling. After explaining the framework and the aims of the study, informed consensus was obtained from all the participants. A semi-structured questionnaire was used to record the following information on the ethnoveterinary uses: (1) informant name, (2) age, (3) ethnicity, (4) date of the interview, (5) geographic location, (6) education level, (7) plant folk name, (8) part of the plant used, (9) disease treated, (10) preparation method, and (11) administration mode. The informants were asked to give their perceptions of the availability of each species as abundant, moderately abundant, and rare. The participants were also encouraged to freely talk about other topics related to pastoralism, including practical aspects of transhumance, conflict resolution, changes over time, and cultural preservation.

Each interview was conducted in the local language and later translated into Portuguese. All interviews adhered to ethical standards as outlined in [122]. Specimens of the mentioned plants were gathered with the help of the participant with the exception of common or cultivated plants. Although quoted during the interviews, some plants were not shown to the researchers and for this reason, they were excluded from the dataset. Vouchers were deposited at the ISCED Herbarium (Lubango, Huila, Angola) except for some samples badly damaged during preparation and/or transport to the Herbarium, which had to be discarded. Identification was carried out by using taxonomic keys and field guides (see information reported in [32]). Taxonomic nomenclature was in accordance with World Flora Online (https://www.worldfloraonline.org/, accessed on 13 September 2023).

### 4.5. Data Analysis

Data collected in the field were entered into a database (Microsoft Excel, 2019), which was organized as a spreadsheet. Each row, considered as a single record, represents a citation, which is a single use reported for a single plant by a single informant [123]. Diseases were grouped into 16 categories according to the symptoms reported during the interviews (Table 3). The experience of veterinary technicians was crucial in translating the symptoms and the folk names of diseases mentioned by the informants into disease categories. The fidelity level was calculated for assessing the specificity of a treatment [124]. Jaccard’s index was used to compare (1) species similarity between our study and studies conducted in other regions of Southern Africa and (2) the usages of plants in the different ethnic groups. This index was corrected for maximum possible values to limit potential artifacts due to the different number of informants interviewed within each group [125]. An abundance index according to the informants’ perception (PAI) was calculated for species having ≥5 citations as follows: ((n_l_ × 1) + (n_m_ × 2) + (n_a_ × 3)/(n_l_ + n_m_ + n_a_) where n_l_, n_m_, and n_a_ represent the number of times each abundance category (low, moderate, and abundant) was assigned to a given species according to the informants’ perception. The relationships between ethnic groups and used categories were examined by a circular plot, using the circlize package (vers. 0.4–2 [126]).

## Figures and Tables

**Figure 1 plants-13-00670-f001:**
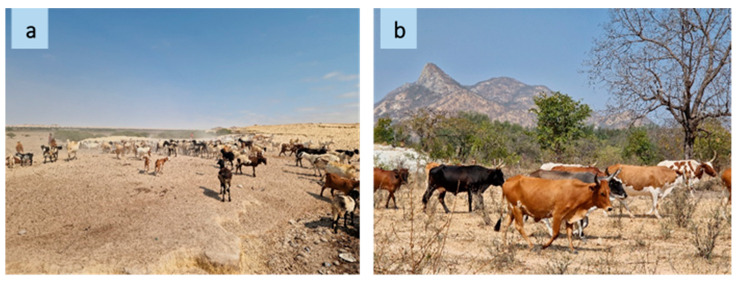
The landscape of a grazing area along transhumance routes ((**a**) Namibe, source: R. Bozzi, 2023; (**b**) Huila, source: R. Bozzi, 2023).

**Figure 2 plants-13-00670-f002:**
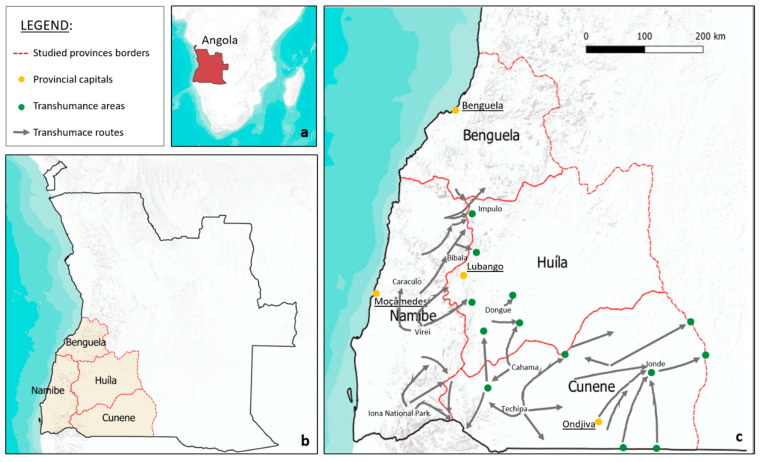
Map of Angola (**a**); provinces where the study occurred (**b**); key transhumance routes (grey arrows) and significant grazing locations (green dots) (**c**).

**Figure 3 plants-13-00670-f003:**
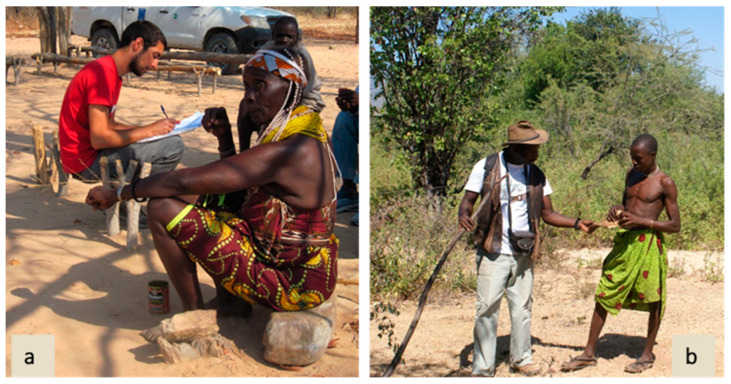
(**a**) The first author interviewing Mumuhuila people about local plants and their veterinary uses (Bibala, source: M. Tonini, 2015). (**b**) A research assistant from the Veterinary Services Institute (ISV) interviewing a shepherd from Bibala (source: P. Bruschi, 2015). The participants of the study consented to the publication of the photos.

**Figure 4 plants-13-00670-f004:**
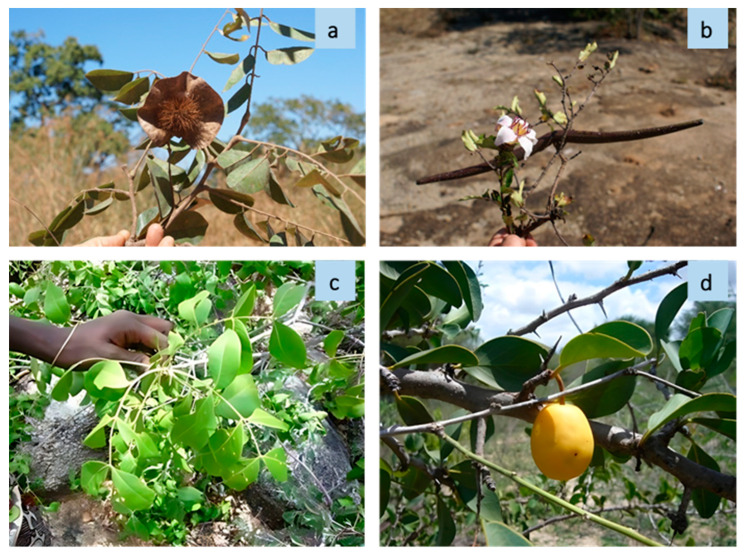
Some useful medicinal plants indicated by the informants: (**a**) *Pterocarpus angolensis* (source: P. Bruschi, Bibala 2015), (**b**) *Pachypodium lealii* (source P. Bruschi, Bibala 2015), (**c**) *Ptaeroxylon obliquum* (source P. Bruschi, Bibala 2015), and (**d**) *Ximenia americana* (source: P. Bruschi, Bibala 2015).

**Figure 5 plants-13-00670-f005:**
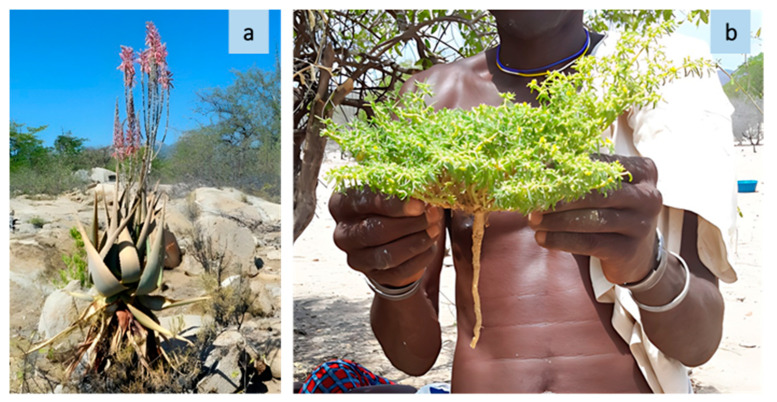
Useful medicinal plants indicated by the informants (**a**) *Aloe littoralis* (source: P. Bruschi, Namibe 2015); (**b**) *Zygophyllum simplex* (source: K. Kamba, Bibala 2014).

**Figure 6 plants-13-00670-f006:**
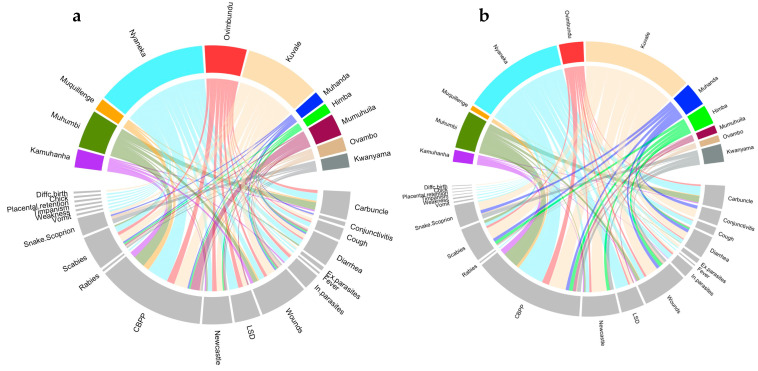
Circular plots showing the relationship between the reported livestock disease and the different ethnic groups. Color bands summarize relationships between each disease and ethnic group (ticks represent the number of species (**a**) and the number of citations (**b**) for each disease and ethnic group, see Table 3).

**Figure 7 plants-13-00670-f007:**
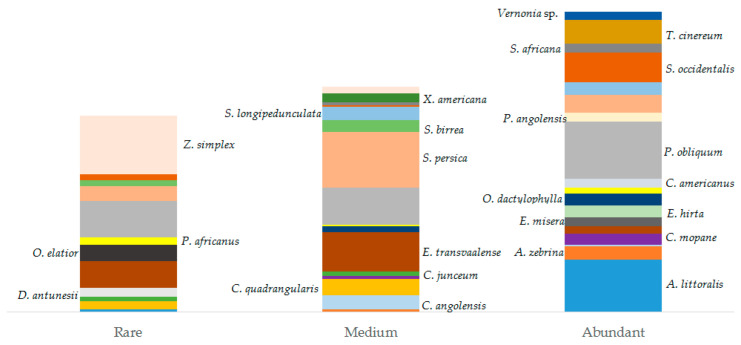
Distribution of species (≥5 citations) abundance according to the informants’ perception. Species were grouped into three categories (*x*-axis): rare, medium, and abundant. Each colored box represents a species and box sizes represent the number of interviewees reporting a certain abundance category for each species. Species can occur in more than one category (e.g., *P. obliquum* is reported in the three abundance categories) because the graph is based on the number of interviewees indicating a certain abundance category for each species. We included the scientific name of the species in the box of species with a higher number of reports in each category.

**Figure 8 plants-13-00670-f008:**
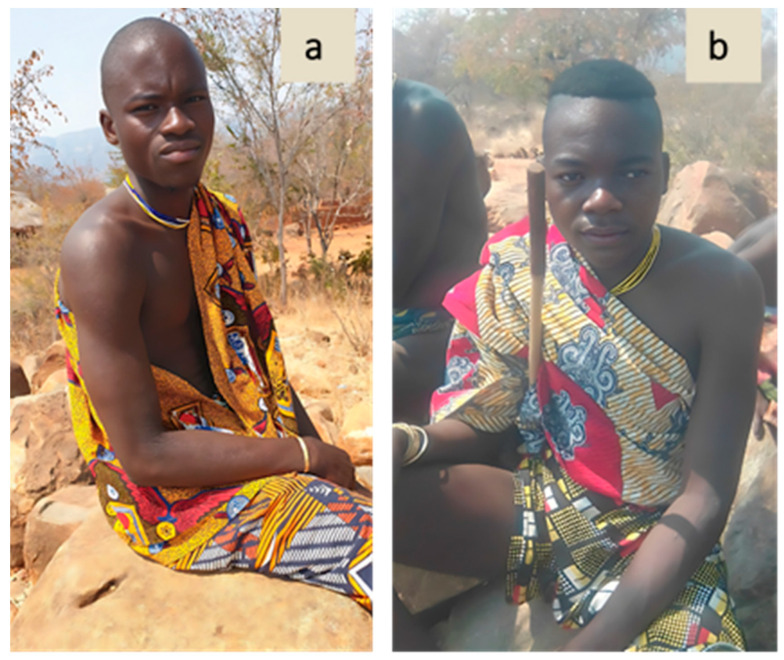
Young shepherds from Kamacuio (**a**) and Bentiaba (**b**) interviewed in Namibe by ISCED technicians in 2023. The participants of the study consented to the publication of the photos.

**Table 1 plants-13-00670-t001:** Informants reporting ethnoveterinary uses in the studied area.

					Ethnolingustic Group														
Province	Municipality	Total Number of Informants	Gender		Ambo		Herero			Nanheca—Humbe						Ovimbundu			
					**Peoples**														
					*KWA*	*OVA*	*KUV*	*HIM*	*MUN*	*MUHA*	*MUHU*	*MUMU*	*MUMG*	*MUQ*	*NHA*	*KIM*	*MUA*	*UMB*	*Unknown*
			Female	Male															
Benguela	Chongoroi	24	4	20															
**Total**		**24**	**4**	**20**	**0**	**0**	**0**	**0**	**0**	**0**	**4**	**0**	**0**	**2**	**0**	**0**	**15**	**3**	**0**
Cunene	Cahama	8	1	7															
	Cuanhama	13	1	12															
	Cuvelai	13	0	13															
	Namacunde	19	7	12															
	Ombaja	9	2	7															
**Total**		**62**	**11**	**51**	**32**	**14**	**0**	**0**	**1**	7	**0**	1	6	0	1	0	0	0	0
Huila	Gambos	21	0	21															
	Humpata	21	0	21															
	Quilengues	24	0	24															
**Total**		**66**	**0**	**66**	**0**	**0**	**0**	**0**	**0**	**0**	20	**0**	**0**	3	39	0	0	4	0
Namibe	Bentiaba	1	1	0															
	Bibala	44	4	40															
	Caitou	4	1	3															
	Camucuio	14	0	14															
	Capangombe	4	0	4															
	Curoca	2	0	2															
	Girul	3	0	3															
	Moçamedes	26	0	26															
	Mutipa	5	2	3															
	Tombwa	30	5	25															
	Virei	26	0	26															
**Total**		**159**	**13**	**146**	**0**	**0**	**46**	**7**	**0**	**0**	**21**	**8**	**0**	**0**	**27**	**1**	**0**	**23**	**26**
**Total over all provinces**		**311**	**28**	**283**	**32**	**14**	**46**	**7**	**1**	**7**	**45**	**9**	**6**	**5**	**67**	**1**	**15**	**30**	**26**

Legend: KWA: Kwanyama; OVA: Ovambo; KUV: Kuvale; HIM: Himba; MUN: Mundimba; MUHA: Muhanda; MUHU: Muhumbi; MUMU: Mumuhuila; MUMB: Mumgabwe; MUQ: Muquilinge; NHA: Nyaneka; KIM: Kimbari; MUA: Muañha; UMB: Umbundu.

**Table 2 plants-13-00670-t002:** List of veterinary taxa used in the studied area.

Species	Voucher	Family	N/I	Local Name	CIT	INF	Used Parts	Desease	Preparation	Administration
*Acacia* sp.		Fabaceae		Muzarere	1	1	B, L, R	CBPP	MAC	OR
*Adansonia digitata* L.	CUA002	Malvaceae	N	Embondeiro	1	1	B, L, R	CBPP	MAC	OR
*Afzelia quanzensis* Welw.	QUI012	Fabaceae	N	Nganga, Muvandji, Movange	3	3	B, FR, L, R	CBPP, diarrhea	MAC, RAW, SP	INH, OR, TP
*Allium sativum* L.	CHO005	Amaryllidaceae	I	Alho	1	1	ST	Newcastle	MAC	OR
*Aloe littoralis* Baker	VIR004 GAM003 HUT002 CHO012 BIB006 CAM009 BIB006	Asphodelaceae	N	Chandala, Chandale, Mandombo, Tchindombo, Tchindombue	46	37	B, FL, L, ST	carbuncle, CBPP, cough, diarrhea, external parasites, internal parasites, Newcastle, scabies, wounds	DEC, SP, MAC, MU, RAW	BA, OR, TP
*Aloe zebrina* Baker	OMB001 TOM001 CUV003 CUV004 CUA003 NAM001	Asphodelaceae	N	Chandola, Chandolo, Endobo	11	11	B, L, ST, WP	CBPP, conjunctivitis, Newcastle, wounds	INF, SP, MAC, RAW	OR, TP
*Balanites angolensis* (Welw.) Welw. ex Mildbr. and Schltr.	VIR002	Zygophyllaceae	N	Mufhimalanda	1	1	ST	snake bite	BUR	INH
*Brachystegia spiciformis* Benth.		Fabaceae	N	Mupanda	1	1		CBPP		
*Brackenridgea areanaria* (De Wild. and T.Durand) N.Robson		Ochnaceae	N	Musenho	3	3	B, R	wounds	POW	TP
*Burkea africana* Hook.	QUI016	Fabaceae	N	Nganga	1	1	B, L	wounds	POW	TP
*Cajanus cajan* (L.) Mill.	CUA005	Fabaceae	I	Handu	1	1	R	CBPP	MAC	OR
*Cannabis sativa* L.	VIR012 CUV005	Cannabaceae	I	Epangwe, Liamba (Canabis)	3	3	L, S	cough, CBPP	INF, MAC	OR
*Capsicum frutescens* L.	TOM002 CHO006 MOÇ003 VIR007	Solanaceae	I	Gindungueiro, Chindungo	47	47	FR, ST	Newcastle, snake bite	MAC, MU, RAW, SP	OR
*Cassia angolensis* Hiern	GAM002 KAH003	Fabaceae	N	Mutangalolo, Mutangalolo/Muhipanhoca	10	7	B	diarrhea, Newcastle, CBPP	MAC	OR
*Cassytha filiformis* L.	CUV006	Lauraceae	N	Muavava	1	1	L, R	CBPP	INF	OR
*Cenchrus americanus* (L.) Morrone R. Br.	CUV006	Poaceae	N	Massango	6	6	S	conjunctivitis	POW	TP
*Cissampelos mucronata* A. Rich	BIB005	Menispermaceae	N	Putchi	1	1	L, R	wounds	POW	TP
*Cissus quadrangularis* L.	HUT004 KAH001 QUI004 BIB004	Vitaceae	N	Eliankopo, Fhindawe, Homengue-Wacandimba, Liancopo	16	11	L, ST, WP	CBPP, caarbuncle, internal parasites, Newcastle	INF, MAC, POW	OR
*Colophospermum mopane* (J.Kirk ex Benth.)	VIR003 MOÇ010 CUA017	Fabaceae	N	Mutuate, Mutuaty, Omuhakwatwa, Oshipeca/Oshihapesha	7	6	B, L, R	CBPP, external parasites, internal parasites, scabies, wounds	MAC, MU, OIN, RAW, SP	OR, TP
*Combretum imberbe* Wawra	VIR020	Combretaceae	N	Mucambi, Mupumpa	2	2	B, R	diarrhea	MAC	OR
*Commiphora mollis* (Oliv.) Engl.		Burseraceae	N	Ngonguela, Ongombe, Ontchitiati	3	1	B, L	CBPP, caarbuncle, LSD		
*Croton gratissimus* var. gratissimus	KAH008 QUI010	Euphorbiaceae	N	Mbango, Ombango, Omumbango, Onambumbu	8	7	B, L, R, ST	CBPP, external parasites, Newcastle, scabies, wounds	BUR, DEC, INF, RAW	INH, OR
*Croton integrifolius* Pax.	CHO007	Euphorbiaceae	N	Katete	2	1	L	CBPP, LSD	INF	OR
*Cussonia angolensis* (Seem.) Hiern		Araliaceae	N	Mupunga	1	1	L	CBPP	INF	OR
*Cyperus articulatus* L.	VIR025	Cyperaceae	N	Tchindaue	2	2	ST	LSD	MAC	OR
*Cyphostemma junceum* (Webb) Desc. Ex Wild and R.B. Drumm	GAM004 CHO002	Vitaceae	N	Combaluva, Ñhamti	8	4	R	CBPP, caarbuncle, diarrhea, scabies, snake bite	MAC, MU	OR, TP
*Dalbergia nitidula* Welw. ex Baker	QUI011 QUI014	Fabaceae	N	Mutona, Omutona	6	6	B, FL, L, R, ST	CBPP, caarbuncle, diarrhea	BUR, DEC, INF	INH, OR
*Dialium englerianum* Henriq.	CUA008	Fabaceae	N	Omfimba	1	1	B, R	diarrhea	MAC	OR
*Dichrostachys cinerea* (L.) Wight and Arn.	KAH005	Fabaceae	N	Enikanica, Omukange	1	1	B	conjunctivitis	POW	TP
*Dicoma antunesii* O.Hoffm.	GAM007	Asteraceae	N	Kaundu	6	2	R, T	CBPP, diarrhea, rabies, snake bite	MAC, MU, POW	OR, TP
*Diospyros kirkii* Hiern		Ebenaceae	N	Nonhande	2	1	B	scabies	SP	OR
*Diospyros mespiliformis* Hochst. ex A.DC.	NAM008	Ebenaceae	N	Omuandi	1	1	FR	CBPP	INF	OR
*Elaeodendron transvaalense* (Burtt Davy) R.H.Archer	MOÇ007 TOM004 VIR006	Celastraceae	N	Mulenkuele	49	49	B, L, R, ST	Newcastle, wounds	AS, BUR, MAC, MU, POW, RAW	OR, TP
*Entandrophragma spicatum* (C.DC.) Sprague	NAM015	Meliaceae	N	Ontako	1	1	B	Newcastle	MAC	OR
*Erlangea misera* S. Moore	CUV010	Asteraceae	N	Mungonga	6	6	R	wounds	POW	TP
*Erythrina abyssinica* DC.	QUI005	Fabaceae	N	Mbanga-Lunda	1	1	B	CBPP	INF	OR
*Erythrophleum africanum* (Benth.) Harms.	CUA014	Fabaceae	N	Ongai	1	1	B	scabies	MAC	BA, OR
*Euclea crispa* (Thunb.) Gürke	CUA010	Ebenaceae	N	Munyime, Onimé	2	1	WP	caarbuncle, cough, scabies	MAC	BA, OR
*Euphorbia hirta* L.	GAM001	Euphorbiaceae	I	Mumpai	8	2	L, R	CBPP, fever, vomit, weakness	MAC	OR
*Fadogia cienkowskii* var. lanceolata Robyns	VIR013	Rubiaceae	N	Chifuaca	2	2	ST	CBPP	SP	OR
*Faidherbia albida* (Delile) A.Chev.		Fabaceae	N	Munhele	1	1	L	wounds	POW	TP
*Ficus* sp		Moraceae		Figueira, Mucuyo	10	7	L	caarbuncle, LSD, CBPP	MU	OR
*Gomphocarpus tomentosum* Burch.	CUA004	Apocynaceae	N	Etampia	1	1	WP	LSD	MAC	OR
*Grewia* sp.	BIB008	Malvaceae		Mumbole	4	2	L, ST	caarbuncle, CBPP, complicated births, skin infection	DEC, MAC, OIN	OR, TP
*Helichrysum benguellense* Hiern	QUI008	Asteraceae	N	Otchitete	2	2	WP	CBPP	INF	OR
*Hyphaene petersiana* Klotzsch ex Mart.	NAM014	Arecaceae		Onlunga	1	1	S	cough	POW	OR
*Ipomoea prismatosyphon* var. prismatosyphon	NAM005	Convulvolaceae	N	Madimba, Ndimbo, Ondime	10	8	B, L	caarbuncle, cough, diarrhea, LSD, Newcastle	MAC	OR
*Jacaranda mimosifolia* D.Don.	MOÇ002	Bignoniaceae	I	Carobi	1	1	L	Newcastle	MAC	OR
*Julbernardia paniculata* (Benth.) Troupin		Fabaceae	N	Mumue, Mwe, Olumwe	2	2	FR	CBPP	RAW	OR
*Landolphia parvifolia* K. Schum.	VIR023	Apocynaceae	N	Mutuangue	4	2	R	snake bite	BUR	INH
*Musa acuminata* Colla	GAM009	Musaceae	I	Bananeira	2	2	WP	internal parasites	RAW	OR
*Nicotiana tabacum* L.	NAM021	Solanaceae	I	Tabaco	1	1	L, ST	snake bite	BUR	INH
*Oldfieldia dactylophylla* (Welw. ex Oliv.) J. Léonard	CAM002 CAM010 VIR009	Picrodendraceae	N	Muipanganga, Thumbo	12	12	B, L, R	CBPP	BUR, SP, MAC	INH, OR
*Opuntia ficus-indica* (L.) Mill.	GAM010 QUI017	Cactaceae	I	Ovimbalimbali, Tabaibeiro, Tabaibo	3	3	L, WP	CBPP, scabies, wounds	SP, MAC, MU	OR, TP
*Otomeria elatior* (A.Rich.) Verdc.	MOÇ008	Rubiaceae	N	Mululu, Nhululu	11	11	B, L, R	diarrhea, scabies	MAC	OR
*Pachycarpus lineolatus* (Decne.) Bullock	KAH002	Apocynaceae	N	Eliko	1	1	WP	wounds	POW	TP
*Pachypodium lealii* Welw.	BIB009	Apocynaceae	N	Tchifuanga	3	2	L, ST	CBPP, wounds	MAC, RAW	OR, TP
*Parinari curatellifolia* Planch. ex Benth.	QUI001	Chrysobalanaceae	N	Ouchau	1	1	B	scabies	MAC	BA
*Pavetta schumanniana* F.Hoffm. ex K.Schum.	QUI007	Rubiaceae	N	Otchingue, Tchingue	4	3	B	CBPP, scabies	INF, MAC	OR, BA
*Peltophorum africanum* Sond.	CUA012 NAM010 OMB005 CUV016 CUA012 CAM006	Fabaceae	N	Mumbungululu, Mupalala, Mupapa, Ompalala, Omupalala	25	18	B, B, L, R	CBPP, conjunctivitis, LSD, scabies, wounded	BUR, CMU, DEC, INF, MAC, POW.	BA, INH, OR, TP
*Pericopsis angolensis* (Baker) Meeuwen	QUI003	Fabaceae	N	Muamba, Ouchamba	1	1	L	wounds	MU	TP
*Philenoptera pallescens* (Welw. ex Baker) Schrire	QUI006	Fabaceae	N	Opandambala	2	2	B	CBPP	INF	OR
*Physalis peruviana* L.		Solanaceae	I	Thepathepa	1	1	L, ST	caarbuncle	BUR	INH
*Psidium guajava* L.	BIB002	Myrtaceae	I	Goiabeira	1	1	L	internal parasites	MAC	OR
*Ptaeroxylon obliquum* (Thunb.) Radlk.	HUT001 MOÇ001 TOM003 CHO008 QUI002 GAM005 KAH004 CHO008 CAM005 CUV015 BIB001 VIR010	Rutaceae	N	Bungululu, Dombo, Lumbungululu, Mbungululo, Mbungululu, Mbungunlunlu, Mumbungululu, Nganga	132	105	B, L, R, ST	caarbuncle, CBPP, Chikungunya virus (CHIKV), cough, diarrhea, internal parasites, LSD, Newcastle, scabies, timpanism, wounds	DEC, INF, MAC, MU, BUR, MU, POW	BA, INH, OR, TP
*Pterocarpus angolensis* DC.	CUV009	Fabaceae	N	Mulilahandi	6	6	B	diarrhea	MAC	OR
*Salvadora persica* L.	VIR001 MOÇ006 TOM006 TOM007	Salvadoraceae	N	Omukambi, Omukanli	59	55	B, L, R	CBPP, external parasites	MAC, POW, RAW	OR, TP
*Sclerocarya birrea* Hochst.	CUA011 CUV008 NAM009 NAM011	Anacardiaceae	N	Muhongo, Omuhongo, Omunghongo, Onghongo, Onhongo	12	12	B, L, ST	caarbuncle, snake bite	BUR, MAC	FUM, TP
*Searsia tenuinervis* (Engl.) Moffett	NAM018 OMB006	Anacardiaceae	N	Ompombo, Opombo	3	3	L, R, ST	CBPP, snake bite	BUR, MAC, MU	INH, OR, TP
*Securidaca longepedunculata* Fresen.	HUT003	Polygalaceae	N	Mussansa, Mussossa, Yanchi	17	13	B, L, S, ST	CBPP, inflammations, internal parasites, placental retention, scabies, wounds	SP, MAC, OIN, POW	MAS, OR, TP
*Senna occidentalis* (L.) Link	CUA015 VIR016 VIR017	Fabaceae	I	Muihanhuca, Muipanhoca, Omuipanyoka, Onty-Weyoca, Onty-Weyoka	26	20	B, L, R, S, ST	CBPP, internal parasites, LsD, Newcastle, snake bite	BUR, INF, MAC	INH, OR
*Senna singueana* (Delile) Lock		Fabaceae	N	Muhanga, Muhangui	2	1	B	LSD	SP	OR
*Solanum lycopersicum* L.	NAM022	Solanaceae	I	Tomateiro	1	1	L	external parasites	MU	EN
*Spirostachys africana* Sond.	CHO009 QUI015 CHO013	Euphorbiaceae	N	Mupapa, Ndjilite, Ndjirite, Ndjiriti, Njiriti	30	21	B, L, R, ST	caarbuncle, CBPP, cough, LSD, Newcastle	BUR, DEC, INF, MAC	BA, INH, OR
*Stachytarpheta mutabilis* (Jacq.) Vahl		Verbenaceae	I	Muhoya	1	1	L	diarrhea	MAC	OR
*Steganotaenia araliacea* Hochst.	QUI009	Apiaceae	N	Epondo	2	2	B	caarbuncle	INF	OR
*Sterculia setigera* Delile	VIR023	Malvaceae	N	Mutengue, Mutengui ya kandimba, Omutengue	4	3	ST	cough, diarrhea, LSD	MAC	OR, TP
*Strychnos henningsi* Gilg.	CAM007	Loganiaceae	N	Ohanaune	1	1	L, ST	caarbuncle	MAC	OR
*Strychnos* sp.	CHO004	Loganiaceae		Muuandai, Muvandji, Nganga, Uganga	6	6	L, ST	wounds	MU, POW, RAW	TP
*Tephrosia vogelii* Hook.f.		Fabaceae	N	Calembi	3	2	L	CBPP, cough, Newcastle	MAC	BA, OR
*Thesium cinereum* A. W. Hill	VIR015	Santalaceae	N	Mkangue	16	16	ST	conjunctivitis	BUR, AS, POW	TP
*Triumfetta* sp	NAM016	Malvaceae		Onti Ombwa	1	1	ST	rabies	BUR, AS	OR
*Vachellia erioloba* (E.Mey.) P.J.H.Hurter	OMB007-OMB008	Fabaceae	N	Omuonde efossa, Omwonde	4	4	B, L, R	CBPP, wounds	MAC, POW	OR, TP
*Vernonia* sp.	CHO003	Asteraceae		Udongo	5	3	FL, L	caarbuncle, CBPP, scabies, wounds	MAC, MU	OR, TP
*Vernonia steetziana* Oliv. and Hiern	OMB004	Asteraceae	N	Okashila	1	1	L, R	snake bite	MU	TP
*Vernoniastrum latifolium* (Steetz) H.Rob.	KAH006	Asteraceae	N	Omumbafi	1	1	B	Newcastle	MAC	OR
*Ximenia americana* L.	BIB003 CUV011 KAH007	Olacaceae	N	Mumpeque, Omupeke	8	8	FR, L	CBPP, scabies, snake bite	MAC, CMU, RAW	OR, TP
*Zea mays* L.		Poaceae	I	Milho	2	2	L, B	CBPP	BUR, INF	OR
*Zygophyllum simplex* L.	MOÇ011 TOM008	Zygophyllaceae	N	Thihele	43	43	B, L, R, ST	scabies	MAC, MU, POW, RAW	OR, TP

Legend. N/I: native/introduced; CIT: number of citations; INF: number of informants; used parts—B: bark; FL: flower; FR: fruit; L: leaves; R: roots; ST: stem; T: tuber; S: seeds; WP: whole plants; preparation—AS: ashes; BUR: burned; CMU: cooked mush; DEC: decoction; INF: infusion; MAC: maceration in water; MU: musk; OIN: ointment; POW: powder; RAW: as raw; SP: sap; administration—BA: bath; EN: enema; FUM: fumigation; INH: inhalation; MAS; massage; OR: oral; TP: topical.

**Table 3 plants-13-00670-t003:** List of treated diseases.

				Ethnic Groups											
	INF	CIT	TAXA	KW	OV	KV	HI	MHA	MHU	MUM	MUN	MUQ	NYA	MUA	UMB
				TAXA											
Carbuncle	37	46	17	2	1	1			7		1	1	5	4	3
CBPP (contagious bovine pleuropneumonia)	190	223	44	5	2	12	1	1	13	7		4	17	4	9
CHICK (chikungunya virus)	1	1	1										1		
Conjunctivitis	25	25	5	1	1	1		1			1				
Cough	13	18	10	1	3	4		1					4		2
Diarrhea	58	60	14	1	1	4	1	1	3	2			6		1
Difficult birth	1	1	1			1									
External parasites	9	9	4	1		1							1		1
Fever	2	2	1										1		
Inflammations	2	2	1										1		
Internal parasites	20	20	8	1		1				1			4	1	2
LSD (lumpy skin disease)	24	40	12	1		4			1	2			6		4
Newcastle	81	86	14	3		5	1		1	1	1		4	2	4
Placenta retention	2	2	2		1								1		
Rabies	3	3	2	1									1		
Scabies	80	83	16	3	1	6	1	1	1	1			5	2	5
Snake/scoprion bite	34	41	12	4	3	3	1	1			2		1	1	
Timpanism	1	1	1										1		
Vomit	2	2	1										1		
Weakness	2	2	1										1		
Wounds	93	95	20	1	1	7	1	1	5	1	1	1	4	3	3
**Total taxa**				**20**	**8**	**30**	**5**	**7**	**24**	**12**	**6**	**6**	**25**	**7**	**18**
**Total taxa (≥3 citations)**				**8**	**7**	**21**	**5**	**7**	**17**	**9**	**3**	**5**	**21**	**5**	**13**
**Total uses**				**25**	**14**	**50**	**6**	**7**	**31**	**15**	**6**	**6**	**65**	**17**	**34**
**Total uses (≥3 citations)**				**4**	**1**	**27**	**5**	**7**	**19**	**9**	**3**	**5**	**31**	**6**	**18**
**Specific ues (≥3 citations)**				**2**	**0**	**6**	**0**	**4**	**2**	**0**	**0**	**0**	**3**	**0**	**0**

Legend. CIT: number of citations; INF: number of informants; TAXA: number of taxa; KW: Kwanhama; OV: Ovambo; KV: Kuvale; HI: Himba; MHA: Muhanda; MHU: Muhumbi; MUM: Mumuhuila; MUN: Mungambwe; MUQ: Muquilengue; NYA: Nyaneka; MUA: Muañha; UMB: Umbundo.

## Data Availability

All data supporting the results of this research are included within the article.

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
