# Peer review of "Preserving Ethnoveterinary Medicine (EVM) along the Transhumance Routes in Southwestern Angola: Synergies between International Cooperation and Academic Researchâ€"

_plants, 2024, doi:10.3390/plants13050670_

Round 1
Reviewer 1 Report
Comments and Suggestions for Authors
Comments to the author:
Binomials must be mentioned with author citation on its first mention
Line 22: mentions the full form for all the abbreviations in its first mention
Lines 29-30: The conclusion part must emphasize the research gap and future scope of the study clearly. Rephrase the sentences.
All the discussed references cited in the text must be mentioned with the author's name.
Check the spelling mistake ‘Bignionaceae’
Page 2 lines 61, 67, 73-75 Rephrase the sentences for clear understanding.
Line 109. Denote whether it is a table or figure for reference without ‘See’
Page 7 line 1. Mention the study area in the title.
Page 7 lines 28&29. Binomials must be italicized
Page 17 lines 11, 18 Rephrase the sentences.
From page 20
Line 9 The only use reported for E. misera is its use against vomiting, then how this plant is categorized as the main plant used for treating wounds.
Line 14 Replace the term remedy.
Mention the abbreviation FL in all the required place.
Line 16 Reference format must be checked with the journal guidelines ‘[56]; [57];’.
Line 19 Mention the used part of Aloe secundiflora, while discussing in detail
Line 20 Check the grammatical error - ‘with scabies’
Line 38 Check the grammatical error - S. occidentalis is a
Line 46 Check the grammatical error – with
Lines 48, 62, 65, 185, 234, 263 Rephrase the mentioned sentences.
Lines 56, 102, 109: (Table A2, Appendix C) is enough ‘See’
Line 76 Kuvale and Nyaneka were the most knowledgeable ethnic groups. Rephrase or remove this statement
Line 84 Remove the statement
Line 97 Check the spelling for livestock
Line 127 Italicize P. obliquum
Line 166 What does the author mean by ‘for a rapid in-field’?
Line 167 State why the crucial approach is required in the same sentence
Lines 209, 213, 249, 269, 272, 294, 309 Colloquial convos with futile sentences, if possible the authors can reframe those sentences.
Line 234 Check the punctuation mark
Line 324 Currency values must be denoted with the equivalency of a standard currency like the dollar.
Lines 327- 331 Remove the sentences
Line 357: Check the grammatical errors - plays crucial; planning and developing
Figure 6: Italicize the binomial names given in the legend. Is the source important here? It can be specified in author's contributions
In Appendix B binomial names must be italicized
Toxicity associated with the plants can be emphasized
Author Response
FIRST REVIEWER
Thank you very much for taking the time to review this manuscript. Please find the detailed responses below each of your comment and the corresponding revisions/corrections highlighted in the re-submitted files.
Comments to the author:
- Binomials must be mentioned with author citation on its first mention
Done
- Line 22: mentions the full form for all the abbreviations in its first mention
Done
- Lines 29-30: The conclusion part must emphasize the research gap and future scope of the study clearly. Rephrase the sentences.
The sentences has been rephrased as follows: Future studies should investigate the opportunity to integrate traditional medicine into mainstream development projects is crucial for decolonising the veterinary sector in Angola.
- All the discussed references cited in the text must be mentioned with the author's name.
Regarding the author’s name and references we’ve followed the Journal guidelines
- Check the spelling mistake ‘Bignionaceae’
Done
- Page 2 lines 61, 67, 73-75 Rephrase the sentences for clear understanding.
The sentences has been rephrased as follows:
61: In addition, drought exacerbates the effects of parasites and infectious diseases caused by reduced immunity associated with poor nutrition and the increased risk of transmission in overcrowding conditions near water and food sources.
73-75: All these limitations make urgent to search for alternative and complementary approaches to heal livestock diseases.
- Line 109. Denote whether it is a table or figure for reference without ‘See’
Done
- Page 7 line 1. Mention the study area in the title.
Done
- Page 7 lines 28&29. Binomials must be italicized
Done
- Page 17 lines 11, 18 Rephrase the sentences.
The sentences has been rephrased as follows:
11: As an example, the fermented buttermilk produced by Herero and Ovambo of northern Namibia and southern Angola plays an important cultural and economic role
18: The higher number of citations (223 citations; 29% of the total number of citations; 44 species; see Figure 4b) regarded the treatment of CBPP (Contagious Bovine Pleuropneumonia, locally known as Cawenha) (Table 3).
From page 20
- Line 9 The only use reported for miserais its use against vomiting, then how this plant is categorized as the main plant used for treating wounds.
The plant has been categorized as for treating wounds in line with our investigation (see the table in text), but in literature has only one mention as for vomiting and fever in children.
However, we’ve rephrased as followed, to make it more clear:
No literature information is available on E. misera as veterinary remedy. The only medicinal use reported in literature for this species is related to vomiting and fever relief in Kwanyama children
- Line 14 Replace the term remedy.
Done
- Mention the abbreviation FL in all the required place.
We’ve reported the abbreviation FL (as well as “citations”) in the first species mentioned in the sentence.
- Line 16 Reference format must be checked with the journal guidelines ‘[56]; [57];’.
Revised according to Journal guidelines
- Line 19 Mention the used part of Aloe secundiflora, while discussing in detail
We’ve reported the part used from Aloe secundiflora, as mentioned in the cited bibliography.
- Line 20 Check the grammatical error - ‘with scabies’
Done
- Line 38 Check the grammatical error - occidentalisis a
- Done
- Line 46 Check the grammatical error – with
- Done
- Lines 48, 62, 65, 185, 234, 263 Rephrase the mentioned sentences.
The sentences has been rephrased as follows:
48: The only traditional medicinal use of T. cinereum reported in literature is for the treatment of human bronchitis in Southern Angola [72]. Other species belonging to the genus Thesium are used to enhance the sight [73].
62: Ptaeroxylon obliquum was reported by 10 ethnic groups, (…)
65: P. obliquum against CBPP was the most (…)
185: (…) the same folk Nyanheca name for Ptaeroxylon obliquum) (Figure 6b
234: Preserving cultural aspects remains a priority but adaptations could ensure the sustainability of practices in the face of changing circumstances.
263: As suggested by [97], the hegemony of Western approach can have a strong impact on the conservation of the indigenous health traditions but at the same time can cause an opposite effect of cultural resistance and revalorization of traditional medicine.
- Lines 56, 102, 109: (Table A2, Appendix C) is enough ‘See’
Done
- Line 76 Kuvale and Nyaneka were the most knowledgeable ethnic groups. Rephrase or remove this statement
- Line 84 Remove the statement
Done
- Line 97 Check the spelling for livestock
Done
- Line 127 Italicize obliquum
Done
- Line 166 What does the author mean by ‘for a rapid in-field’?
We’ve removed “in-field”
- Line 167 State why the crucial approach is required in the same sentence
We’ve explained later in text.
- Lines 209, 213, 249, 269, 272, 294, 309 Colloquial convos with futile sentences, if possible the authors can reframe those sentences.
The sentences has been reported as from the interviewed people, very much appreciated by reviewer n.3 (he appreciated that “the authors use quotes from people…”)
- Line 234 Check the punctuation mark
Done
- Line 324 Currency values must be denoted with the equivalency of a standard currency like the dollar.
The exchange rate in Angola is highly variable during 12 months, thus it makes very difficult to use a correspondence with a standard currency.
- Lines 327- 331 Remove the sentences
Done
- Line 357: Check the grammatical errors - plays crucial; planning and developing
Done
- Figure 6: Italicize the binomial names given in the legend. Is the source important here? It can be specified in author's contributions
- In Appendix B binomial names must be italicized
Done
- Toxicity associated with the plants can be emphasized
We added a sentence in the Conclusions about the related commented:
Further research is needed for a deeper understanding of the pharmacological activity, therapeutic use and possible toxic effects of the less known mentioned species.
Reviewer 2 Report
Comments and Suggestions for Authors
The text offers a balanced and insightful perspective on the importance of preserving and integrating traditional medical practices within broader healthcare frameworks in Africa. A very good job.

- I suggest carefully rereading the text. For example, in the abstract.
- "thransumance" should be corrected to "transhumance."
- "CBPP" should be spelled out as "Contagious Bovine Pleuropneumonia" on first mention.
- "preservetion" should be corrected to "preservation."
... many other instances within the body of the text.
Author Response
SECOND REVIEWER
Thank you very much for taking the time to review this manuscript and for the insightful comments and constructive feedback provided on our article. We will carefully consider your suggestions in our future research.
Reviewer 3 Report
Comments and Suggestions for Authors
I read your manuscript with great interest. It was very pleasant news for me that such a presidential order has been issued in Angola (Presidential Decree No. 253/20).
The methods and analysis of the manuscript were well described. The results are reliable. The reviewer has only formal remarks.
I loved that the authors used quotes from people and also that the authors used photos, both supplemented the text with very important information. Photos also pass to readers what cannot be described in words. Hence the reviewer's first request, please put the date when they were taken and who took them with the photos. And if possible, also write the place where these pictures were taken.
Another issue requiring technical improvement is the reflection of plant names. In the main text, it is not necessary to add the author's name after the plant names, e.g. page 7 lines 3 to 6, and in the table, the plant names are currently upright but should be in italics (e.g. Table A2.).
I wish the authors success and strength for doing a great job.
P.S. Even the most important scientific journals have now begun to emphasize the importance of native wisdom in their editorials: Kimmerer, R. W., & Artelle, K. A. (2024). Time to support Indigenous science. Science, 383(6680), 243-243.
Author Response
THIRD REVIEWER
Thank you very much for taking the time to review this manuscript and for the constructive feedback provided on our article. Please find the responses below and the corresponding revisions/corrections highlighted in the re-submitted file.
Comments to the author:
- Photos also pass to readers what cannot be described in words. Hence the reviewer's first request, please put the date when they were taken and who took them with the photos. And if possible, also write the place where these pictures were taken.
Done
- Another issue requiring technical improvement is the reflection of plant names. In the main text, it is not necessary to add the author's name after the plant names, e.g. page 7 lines 3 to 6, and in the table (…)
As suggested by the first reviewer (Binomials must be mentioned with author citation on its first mention), we have reported the author’s name after the plant name in first mention. As for page 7, lines 3 to 6, we’ve reported the author’s name since the species are not listed in the table.
- "thransumance" should be corrected to "transhumance."
Done
- "CBPP" should be spelled out as "Contagious Bovine Pleuropneumonia" on first mention.
Done
- "preservetion" should be corrected to "preservation."
Done
Round 2
Reviewer 1 Report
Comments and Suggestions for Authors
I wish to express my sincere appreciation to the authors for their revised manuscript, diligent focus on details, and adept handling of the issues raised during the initial review. Their efforts have resulted in substantial improvements, significantly elevating the quality and clarity of the manuscript. It is suggested to authors that, while responding to the reviewers and editorial comments, the authors should mention the page number/line number for the amendments made by them for a clear understanding of the revision.